# Subgrouping and TargetEd Exercise pRogrammes for knee and hip OsteoArthritis (STEER OA): a systematic review update and individual participant data meta-analysis protocol

Melanie A Holden,[1] Danielle L Burke,[1] Jos Runhaar,[2] Danielle van Der Windt,[1] Richard D Riley,[1] Krysia Dziedzic,[1] Amardeep Legha,[1] Amy L Evans,[1] J Haxby Abbott,[3] Kristin Baker,[4] Jenny Brown,[5] Kim L Bennell,[6] Daniël Bossen,[7,8] Lucie Brosseau,[9] Kanda Chaipinyo,[10] Robin Christensen,[11] Tom Cochrane,[12] Mariette de Rooij,[13] Michael Doherty,[14] Helen P French,[15] Sheila Hickson,[5] Rana S Hinman,[6] Marijke Hopman-Rock,[16,17] Michael V Hurley,[18,19] Carol Ingram,[5] Jesper Knoop,[13] Inga Krauss,[20] Chris McCarthy,[21] Stephen P Messier,[22] Donald L Patrick,[23] Nilay Sahin,[24] Laura A Talbot,[25] Robert Taylor,[5] Carolien H Teirlinck,[2] Marienke van Middelkoop,[2] Christine Walker,[5] Nadine E Foster,[1] in collaboration with the OA Trial Bank

For numbered affiliations see end of article.

**Correspondence to**
Dr Melanie A Holden;
m.holden@keele.ac.uk

## ABSTRACT

**Introduction** Knee and hip osteoarthritis (OA) is a leading cause of disability worldwide. Therapeutic exercise is a recommended core treatment for people with knee and hip OA, however, the observed effect sizes for reducing pain and improving physical function are small to moderate. This may be due to insufficient targeting of exercise to subgroups of people who are most likely to respond and/or suboptimal content of exercise programmes. This study aims to identify: (1) subgroups of people with knee and hip OA that do/do not respond to therapeutic exercise and to different types of exercise and (2) mediators of the effect of therapeutic exercise for reducing pain and improving physical function. This will enable optimal targeting and refining the content of future exercise interventions.

**Methods and analysis** Systematic review and individual participant data meta-analyses. A previous comprehensive systematic review will be updated to identify randomised controlled trials that compare the effects of therapeutic exercise for people with knee and hip OA on pain and physical function to a non-exercise control. Lead authors of eligible trials will be invited to share individual participant data. Trial-level and participant-level characteristics (for baseline variables and outcomes) of included studies will be summarised. Meta-analyses will use a two-stage approach, where effect estimates are obtained for each trial and then synthesised using a random effects model (to account for heterogeneity). All analyses will be on an intention-to-treat principle and all summary meta-analysis estimates will be reported as standardised mean differences with 95% CI.

### Strength and limitation of this study

► This will be the first study in the field of therapeutic exercise and osteoarthritis to combine individual participant data from existing randomised controlled trials.

► Combining individual participant data from existing trials will increase the power to identify who benefits most from therapeutic exercise, and to identify underlying mechanisms of action.

► Individual participant data meta-analyses facilitates standardised analyses across studies, allows direct derivation of desired information independent of significance or reporting, enables subgroup effects and interactions (differences in effects between subgroups) to be examined, and may provide more outcomes than were considered in a single original publication.

► A disadvantage to completing individual participant data meta-analyses is the time required to complete them, including obtaining, checking and combining the data.

**Ethics and dissemination** Research ethical or governance approval is exempt as no new data are being collected and no identifiable participant information will be shared. Findings will be disseminated via national and international conferences, publication in peer-reviewed journals and summaries posted on websites accessed by the public and clinicians.
**PROSPERO registration number** CRD42017054049.

## INTRODUCTION

Osteoarthritis (OA) can be defined as a clinical syndrome of joint pain accompanied by varying degrees of functional limitation and reduced quality of life.[1] OA, particularly of the knee and hip, is one of the leading causes of disability worldwide, with an estimated global age-standardised prevalence of 3.8% (95% CI 3.6% to 4.1%) for knee OA and 0.85% (95% CI 0.74% to 1.02%) for hip OA.[2] The burden of OA will increase as the population ages and the prevalence of obesity rises.[2 3]

No cure currently exists for OA and as such treatment strategies aim to reduce pain and improve physical function, and enhance quality of life.[4] Clinical guidelines, including the National Institute for Health and Care Excellence OA guidelines,[1] consistently recommend therapeutic exercise as a core treatment for people with knee and hip OA.[5 6] Therapeutic exercise involves participation in physical activity that is planned, structured, repetitive and purposeful for the improvement or maintenance of a specific health condition such as OA.[7] It encompasses general aerobic exercise, strengthening, flexibility, balance or body-region specific exercises.[7] Although both general (aerobic) exercise and strengthening exercise are recommended for people with knee and hip OA, limited information is available on the optimal prescription of therapeutic exercise (eg, the optimal type, dose, intensity and setting of exercise and how best to progress exercise).[1] Numerous systematic reviews and meta-analyses support the role of therapeutic exercise for knee and hip OA, consistently demonstrating that it can reduce pain and improve physical function.[8–10] However, results from randomised controlled trials (RCTs) show the observed effect sizes from exercise interventions are small to moderate, can decline over time, and only approximately 50% of participants achieve a clinically important treatment response.[11–13] The modest average benefits obtained from therapeutic exercise could be due to the inclusion of subgroups of people who are unlikely to benefit from exercise, and thus the overall effect is closer to the null than if the trial had been solely undertaken in those that are likely to benefit.[4] Better targeting of exercise treatments for people with knee and hip OA could potentially lead to improved treatment effects and patient outcomes, as well as more efficient use of healthcare services, in a similar way as demonstrated for low back pain.[14 15] Such an approach requires the identification of subgroups who are likely to respond better to therapeutic exercise than others.

Little previous research has examined whether outcomes from exercise for OA vary for subgroups of people defined by individual-level characteristics (treatment effect moderators).[16] Exploratory secondary analyses of some RCTs suggest a range of potential moderators of the effects of exercise, including age,[17] sex,[18] obesity,[19] pain severity and duration,[17 18] functional ability,[18] strength,[20 21] knee malalignment,[19 20] severity of joint damage,[22] anxiety and depression[18] and treatment outcome expectations.[23] However, post hoc analyses

have low statistical power to detect significant subgroup effects, and are at high risk of yielding spurious findings due to multiple testing.[24] Although these exploratory subgroup analyses are inconclusive, they add credence to the hypothesis that not all people with knee and hip OA respond similarly to exercise, with variability in effects related to individual-level characteristics.

Modest average benefits of therapeutic exercise in people with knee and hip OA may additionally be explained by suboptimal content of exercise programmes. Systematic reviews have identified various characteristics of exercise programmes that appear to be associated with larger effects, but with conflicting results.[10 25] Treatment mediators (causal links between treatment and outcome[16]) of therapeutic exercise on OA symptoms are largely unknown, making it difficult to design therapeutic exercise programmes for optimal effects on symptoms. If true mediators were identified and targeted, the positive effects of therapeutic exercise may be improved. Increased muscle strength, decreased extension deficits and improved proprioception have been identified as potential working mechanisms for the positive effect of therapeutic exercise for knee OA, and increased muscle strength for hip OA.[26] However, meta-analyses at the study-level (using aggregated study results)[26] have been prone to study-level confounding regarding the identification of individual-level effects.[25 26]

The investigation of individual response to exercise treatment, and the identification of factors that may cause differential response to such treatment or to particular components of exercise therapy, requires individual-level data analysis. To our knowledge, this type of trial data pooling and analysis has not yet been completed in the field of therapeutic exercise and OA. Although several systematic reviews are available,[8–10] none use individual participant data (IPD). Given there are now over 60 RCTs of exercise for knee and hip OA,[10] it is timely to combine IPD from these existing trials. This will increase the power to identify who benefits most from therapeutic exercise, and to identify underlying mechanisms of action.[27] IPD meta-analysis facilitates standardised analyses across studies, allows direct derivation of desired information independent of significance or reporting, enables subgroup effects and interactions (differences in effects between subgroups) to be examined, and may provide longer follow-up, more participants and more outcomes than were considered in the original publication.[27] Therefore, IPD meta-analyses are potentially more reliable than traditional aggregate data meta-analyses for the identification of treatment effect moderators, may lead to different conclusions and may produce more clinically relevant results.[27]

## AIM

To identify (1) subgroups of people with knee and hip OA that respond/do not respond to therapeutic exercise and to different types of exercise (effect moderators) and

(2) mediators of the effect of therapeutic exercise for reducing pain and improving physical function to facilitate better targeting of future exercise interventions and refine exercise programme content.

## Specific analytic objectives

1. Determine the short-term (12 weeks), medium-term (6 months) and long-term (1 year) overall effects of therapeutic exercise on pain and physical function for people with knee and hip OA compared with a non-exercise control.
2. Determine which study-level characteristics of therapeutic exercise interventions are associated with improved overall effects on pain and physical function in people with knee and hip OA, including the type, intensity, duration, setting and deliverer of exercise.
3. Identify individual-level characteristics of people with knee and hip OA that are associated with the short-term, medium-term and long-term effects of therapeutic exercise on pain and physical function.
4. Identify individual-level characteristics of people with knee and hip OA that are associated with the effects of different characteristics of therapeutic exercise interventions; including the type, intensity, duration, setting and deliverer of exercise.
5. Investigate the effect estimates for objectives 1–4 in subgroups of people with only knee OA and only hip OA to examine whether they differ by joint site.
6. Evaluate the mediating effects of muscle strength (for people with knee and hip OA), proprioception (for people with knee OA) and extension deficits (for people with knee OA) in the association between therapeutic exercise and pain and physical function. The effects of individual and combined mediators will be explored.

## METHODS AND ANALYSIS

We will update a previous systematic review[10] to identify relevant RCTs and after agreement from trial leads, undertake an IPD meta-analysis. Our systematic review and meta-analysis will be completed in accordance with methods advocated by the Cochrane IPD meta-analysis group,[28 29] and reported according to Preferred Reporting Items for Systematic Reviews and Meta-Analyses (PRISMA)-IPD guidance.[30] Five members of the Research User Group at the Research Institute of Primary Care and Health Sciences, Keele University, have formed a Patient and Public Involvement and Engagement (PPIE) working group for this study. In line with INVOLVE[31] there will be PPIE involvement at every stage of the project. We will work in collaboration with the OA Trial Bank (www.oatrialbank.com), an initiative established in 2010 to collect and analyse IPD of published RCTs in OA.[32 33] The final IPD database will be deposited with the OA Trial Bank for the benefit of the wider OA research community.

### Phase 1: trial identification

We previously conducted a systematic review that identified 60 RCTs of exercise interventions for people with knee and hip OA that are relevant for inclusion within this study.[10] We will update this review. The search strategy previously developed will be re-run from the date of the previous search (March 2012) in the following electronic databases: Medline, EMBASE, Cumulative Index to Nursing and Allied Health Literature, Association for Management Education and Development, Health Management Information Consortium, Cochrane Database of Systematic Reviews, Cochrane Controlled Clinical Trials, Database of Reviews of Effectiveness, National Health Service Economic Evaluations Database and Web of Science. Bibliographies of relevant review articles and included articles will be examined for additional potentially relevant trials. There will be no language restriction. Full search strategies for Medline and Embase are shown in online supplementary appendix 1.

### Study selection

Full details of the study selection criteria are listed in table 1. In summary, we will evaluate RCTs against the following inclusion criteria.

#### Study population

Adults aged ≥45 years with knee or hip OA; *intervention:* any land-based or water-based therapeutic exercise intervention regardless of content, duration, frequency or intensity; *comparator:* other forms of exercise or no exercise control group;

#### Outcome measure

At least one measure of self-reported pain or physical function;

#### Study design

RCTs. Trials will be excluded if they concern preoperative or postoperative therapeutic exercise, when exercise is combined with interventions other than advice/education/self-management/motivational techniques (meaning treatment effects cannot clearly be attributed to the exercise), or if intervention groups receive identical therapeutic exercise interventions. Titles and abstracts of identified studies and subsequently full papers will be independently reviewed by two reviewers. A third reviewer will be consulted to resolve disagreements, if necessary.

### Extraction of aggregate data

For each included trial, details on design, sample size, population characteristics (knee OA, hip OA or mixed), interventions (type, duration and exercise deliverer), comparator, candidate baseline variables (potential treatment moderators and mediators) and outcome assessment (type of outcome measure and length of follow-up) will be extracted and summarised into tables. Two reviewers will independently extract outcome data on self-report pain and/or physical function at time points nearest to 12 weeks, 6 months and 12 months postrandomisation.

**Table 1** Inclusion/exclusion criteria

|  | Inclusion criteria | Exclusion criteria |
|---|---|---|
| Population | ▶ Knee and/or hip pain in adults aged ≥45 years (mean age >45 years)<br>▶ Knee and/or hip OA diagnosed by X-ray<br>▶ Knee and/or hip OA diagnosed according to clinical criteria<br>▶ Knee and/or hip OA diagnosed by healthcare professional<br>▶ Self-reported knee and/or hip OA<br><br>NB: If population is mixed (eg, OA and RA, include if over 50% of participants have OA | ▶ Knee and/or hip pain attributable to conditions other than OA<br>▶ Non-musculoskeletal conditions<br>▶ RA/other defined inflammatory rheumatological problems<br>▶ Preoperative patients (people on waiting-lists for knee/hip surgery, including total joint replacement)<br>▶ Postoperative patients (immediately following knee/hip surgery, including total joint replacement)<br>▶ People with 'patellofemoral pain syndrome' (overall a different problem to 'OA')<br>▶ Animal-based studies<br>▶ Studies of children |
| Intervention | ▶ Any therapeutic exercise intervention (land or water based), regardless of content, duration, frequency or intensity | ▶ Non-exercise interventions<br>▶ Advice only to exercise or increase physical activity, including within wider OA self-management programmes<br>▶ Exercise or physical activity that was not specifically applied to improve OA symptoms and function<br>▶ Exercise combined with treatment modalities other than advice/education/self-management/motivational techniques)<br>▶ Preoperative/postoperative exercise therapy, that is, exercise immediately before, or following knee/hip surgery |
| Comparator | ▶ Other forms of exercise (ie, different type, duration, frequency or intensity of exercise if sufficiently different from the intervention arm)<br>▶ No exercise control group (including usual care, waiting list, placebo, attention control or no treatment)<br>▶ Sham treatment (eg, sham ultrasound) | ▶ If intervention groups receive identical therapeutic exercise interventions (ie, no contrast existing between the intervention groups)<br>▶ If the comparator is a different intervention other than usual care, waiting list, placebo, attention control or no treatment (eg, manual therapy, ultrasound, intra-articular injection, opioids, weight loss, etc) |
| Outcome measure | ▶ Any self-reported measure of pain and/or physical function | ▶ No measure of self-reported pain and/or physical function |
| Study design | ▶ RCT<br>▶ Quasi-RCT (where the method of allocation is known, but is not considered strictly random, eg, alternation, date of birth and medical record number) | ▶ Non-RCT study design<br>▶ Other study designs for example, surveys, observational studies, pre-experiments and postexperiments (without a control group), qualitative studies<br>▶ Systematic reviews<br>▶ RCT protocols |

OA, osteo arthritis; RA, rheumatoid arthritis; RCT, randomised controlled trial.

Two reviewers will independently classify the exercise interventions of included trials, based on the following a priori defined characteristics:

*Frequency of exercise:* number of exercise sessions per week.

*Intensity of exercise:* low, moderate or high intensity (based on published information regarding target heart rate (<50% of maximum heart rate (MHR)=low intensity, 50%–70% MHR=moderate intensity, >70%–85% MHR=high intensity) or metabolic equivalent (MET) score (where heart rate information is unavailable) (MET score of <3= low intensity, MET 3–6=moderate intensity, MET >6=high intensity[34 35]; low or high impact (categorised based on the likely amount of compressive load and

whether both feet were intermittently off the ground. For example, cycling, swimming and walking=low impact; jogging, running and jumping=high impact).[35]

*Type of exercise:* predominantly strengthening (eg, quadriceps strengthening); predominantly general (aerobic) (eg, walking and swimming); predominantly mind–body (eg, yoga and tai-chi)[36] ; mixed. As many trials are likely to include predominantly strengthening, we will classify subsets of predominantly non-weightbearing/open kinetic chain strengthening exercise versus predominantly weightbearing/closed kinetic chain strengthening exercise.

*Duration of exercise programme:* short (less than 6 weeks) or longer durations of up to 12 weeks, and over 12 weeks;

total number of exercise sessions; booster sessions or no booster sessions.

*Setting of exercise:* group, individual or mixed; supervised in clinic, completed at home or mixed; face-to-face, remote exercise instruction or mixed.

*Exercise deliverer:* healthcare professionals, exercise specialists, peer or lay-led or mixed.

## Phase 2: collection, checking and standardising individual participant data

In collaboration with, and following the procedures of, the OA Trial Bank, we will contact lead authors of identified trials to inform them about the study and invite them to share IPD. If the updated systematic review yields a large additional number of RCTs suitable for inclusion, and many authors are willing to share IPD, we may examine the likely power of the IPD meta-analysis accordingly to the trials promising their IPD using a simulation-based approach.[37] The collection, cleaning and harmonisation of IPD is often resource intensive,[28 38] and therefore the power calculation will inform how much IPD is required in order to obtain sufficient power (eg, 80%) to evaluate our key objectives. If IPD is ultimately sought from a subset of studies, this will be based on study quality, sample size and improvement to power, and independent of effect size to avoid selection bias.[39]

Once a data sharing agreement is in situ, datasets will be accepted in any form, provided all data are anonymised and variables and categories are adequately labelled in English. To ensure accurate pooling of data, each dataset will be converted to a common format and variables renamed in a consistent manner.

### Variables of interest

IPD to be obtained (where available) will include the following:

### Baseline measures
#### Sociodemographic factors

Age, sex, comorbidities, frailty, fatigue/vitality, quality of life, body mass index, baseline physical activity level, previous lower limb injury, work status (working yes/no), manual versus non-manual work, previous physical load, family history of OA, socioeconomic status (education), social support, currently receiving other treatment, smoking status, motivation to exercise and previous knee injury/trauma.

#### Psychological factors

Anxiety/depression, self-efficacy, outcome expectations. Disease characteristics: pain location, pain elsewhere, pain severity, pain duration, central pain sensitisation, pain bothersomeness, physical function, stage of OA (early vs established OA[40]), radiographic joint structure, evidence of synovitis and bone marrow lesions from MRIs and patellofemoral OA damage.

#### Biomechanical factors

Proprioception, static/dynamic alignment, strength of hip and lower limb musculature, range of movement, leg length discrepancy and developmental hip abnormalities/deformities.

#### Outcome measures

All self-report pain and physical function outcome data at time-points nearest to 12 weeks (short-term), 6 months (mid-term) and 1 year (long-term) postrandomisation. If more than one measure of self-reported pain and physical function are reported, we will chose the highest in the hierarchy of outcome measures, as recommended by the Cochrane Musculoskeletal Review Group.[41] For pain these are: (1) pain overall; (2) pain on walking; (3) Western Ontario and McMaster Universities Osteoarthritis Index (WOMAC) pain subscale; (4) pain on activities other than walking; (5) WOMAC global scale; (6) Lequesne osteoarthritis index global score; (7) other algofunctional scale; (8) patient's global assessment; (9) physician's global assessment (10) other outcome and (11) no continuous outcome reported. For physical function, these are: (1) global disability score; (2) walking disability; (3) WOMAC disability subscore; (4) composite disability scores other than WOMAC; (5) disability other than walking; (6) WOMAC global scale; (7) Lequesne osteoarthritis index global score; (8) other algofunctional scale. Additionally, strength of hip and lower limb musculature, range of movement (total range of motion, maximal flexion, maximal extension and extension deficit), and proprioception measures will be obtained during and directly postintervention.

### Data quality assurance

We will evaluate the IPD from each trial to ensure the ranges of included variables are reasonable, and missing data will be checked against the original trial report. We will attempt to re-produce the results included in each initial trial publication, including baseline characteristics and self-reported pain and physical function at a time point nearest to 12 weeks, 6 months and 1 year postrandomisation. Discrepancies or missing information will be discussed and clarified with original trial authors. Where discrepancies cannot be explained the trial data will be excluded from the analysis. Individual trial datasets will be combined to form a new master dataset with a variable added to indicate the original trial.

### Assessment of risk of bias

We will use the Cochrane collaboration's tool for assessing risk of bias, based on publications of the included trials.[42] Two researchers will independently grade risk of bias (unclear, high or low risk of bias) based on sequence generation, allocation concealment, blinding of outcome assessor, incomplete outcome data and selective outcome reporting. Trial design, conduct and analysis methods will be clarified with principal investigators. Additionally, IPD will be directly checked for key potential biases, including

whether baseline participant characteristics are balanced by arm. It will also be checked to ensure that data on all or as many randomised participants as possible are included, and any additional relevant outcome data from trials will be obtained.

## Part 3: statistical analyses

We will describe trial-level and participant-level characteristics (for baseline variables and outcomes) of included studies and examine if RCTs for which IPD are obtained are representative of the full set of existing RCTs by comparing key study characteristics, for example, country of origin and sample size. All meta-analyses, apart from the mediation analyses (objective 6 below), will use a two-stage approach, where each trial is analysed separately in the first stage (which accounts for clustering of participants within trials) to produce effect estimates of interest, which are then synthesised in the second stage to produce summary meta-analysis results based on a random effects model (to account for between-trial heterogeneity).[26] Analyses will be on an intention-to-treat principle and all summary estimates will be reported with 95% CI and P values, with approaches such as Hartung-Knapp used to account for uncertainty of variance estimates.[43 44] Under a 'missing-at-random' assumption, individuals with partially missing outcome data will be included in analyses without imputation using a longitudinal data meta-analysis framework. If there is a considerable amount of missing baseline data (>5% of patients have one or more missing values) for particular variables of interest (such as potential individual-level effect moderators) this will be handled using within-study multiple imputation[45] and Rubin's rule to estimate effects from imputed datasets.[46] All analyses will be carried out using Stata 14.1[47] or SAS 9.3.[48]

### Objective 1

For the meta-analysis to estimate an overall intervention effect (at 12 weeks, 6 months and 1 year) for self-reported pain and physical function, all available comparisons will be grouped into any exercise intervention versus a non-exercise control. Most outcomes will be continuous, so linear regression models will be fitted in the first stage. Longitudinal models will be used to account for the correlation between outcome values at the multiple time-points.[49] For each trial, the model will include baseline pain/physical function, treatment, time and treatment by time interaction terms. The second stage requires a multivariate meta-analysis framework, which jointly syntheses the treatment effect estimates from the multiple time-points across trials.[50] Given the likely heterogeneity in the intervention effects across trials (eg, due to variability in patient characteristics), we will assume a multivariate random-effects meta-analysis model to estimate the summary results of interest using restricted maximum likelihood estimation. Heterogeneity in treatment effects across trials will be summarised by the estimated between-trial variance ($\tau^2$) and multivariate $I^2$ statistics.

### Objective 2

To determine which characteristics of exercise are associated with differences in overall effects, the meta-analysis approach in objective 1 will be repeated for particular subgroups of exercise interventions, including types, intensities, duration, setting or deliverer of exercise. To formally evaluate (although indirect) differences between exercise subgroups, meta-regression will be used. Trials comparing two different forms of exercise interventions will also be summarised, as these give direct (within-trial) information, which is preferable to indirect information. As appropriate, a sensitivity analysis will be performed to include direct and indirect evidence in one large (network) meta-analysis model.

### Objectives 3 and 4

The IPD will be further analysed to examine treatment effect modification at the patient level, where individual patient characteristics are associated with differences in response to exercise. The models fitted in each trial will additionally include interaction terms between the intervention and patient-level covariates of interest to test effect modification. The interaction effect estimates at each time-point will then be pooled across trials using a multivariate random-effects meta-analysis. Covariates will be mean centred to aid the translation of results. Analyses of different exercise interventions (objective 4) will also be extended to examine treatment–covariate interactions in the same way, and identify subgroups of individuals likely to benefit most from specific types, intensities, duration, setting and deliverer of exercise.

### Objective 5

The analyses described for objectives 1–4 will be repeated in subgroups of people with only knee OA and only hip OA to examine whether the effect estimates differ by joint site.

### Objective 6

As depicted in figure 1A, in a mediation analysis an exposure can affect the outcome either through the mediator (E→M→O) or through other pathways (E→O). Using the counterfactual approach,[51] the *total effect* of the exposure (exercise therapy) on the outcome (pain/physical function) through both pathways is determined. This effect can be decomposed into two components: the *direct effect* and the *indirect effect*. The *direct effect* refers to the causal pathway by which exercise therapy has an effect on pain/physical function not through the mediator. The *indirect effect* refers to the effect of exercise therapy that operates solely through the mediator under investigation. The counterfactual approach also allows for multiple mediators (figure 1B).[51] Given the fact that in this approach the *total effect* can be decomposed into the *direct* and *indirect effects*, the percentage mediated by the mediator(s) can be calculated as an estimation of their importance.

In an one-stage approach, first the effect of the intervention (*a*) on the outcome (*Y*) will be determined,

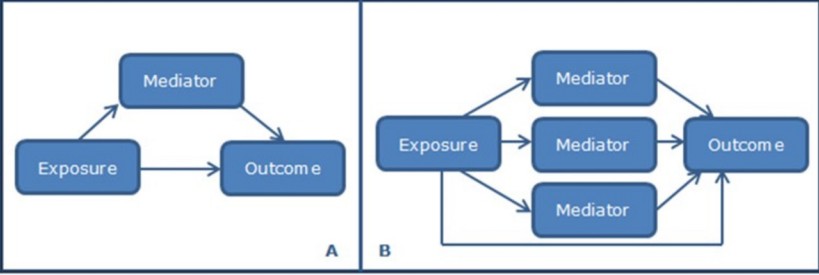

**Figure 1** Causal pathway of potential mediators: (A) single mediator and (B) multiple mediators.

controlling for the mediator ($m$) under investigation and potential mediator-outcome confounders ($c$), using this model:

$$E[Y|a, m, c] = \theta_0 + \theta_1 a + \theta_2 m + \theta_3 am + \theta_4 c$$

Next, the effect of the intervention on the mediator is determined, using this model:

$$E[M|a, c] = \beta_0 + \beta_1 a + \beta_2 c$$

A single covariate will be added to both regression models to indicate each study, in order to adjust for possible residual confounding by study differences. Using these models, where $\theta_i$ and $\beta_i$ are the regression coefficients, the natural direct effect (NDE) and natural indirect effect (NIE) are defined as:

$$NDE = \{\theta_1 + \theta_3 (\beta_0 + \beta_1 a + \beta_2 c)\}$$

NIE = $(\theta_2 \beta_1 + \theta_3 \beta_1 a)$ and the total effect (TE) is equal to the sum of NDE and NIE.[49] Hence, the percentage mediated will be calculated by dividing NIE by TE and multiply this by 100%.

For knee OA, the mediating effect of upper leg muscle strength, extension deficits and proprioception will initially be analysed separately, using all data available for the IPD data meta-analysis. The effect of each potential mediator will then be evaluated adjusting for the other mediators in a multi-moderator model (figure 1B). In the latter, a separate linear regression model will be calculated for each mediator[51] and the NIE, hence the percentage mediated, of each mediator can be calculated.

For hip OA, only the effect of muscle strength will be evaluated since this was the only factor indicated as a potential mediator in a systematic review.[26] In all analyses, the mediator will be defined as the absolute change from preintervention to postintervention. Therefore, the outcome measures that will be used will be those measured as close as possible to the end of the intervention period and to the measurement of the mediator(s) under investigation. All analyses will be run with and without stratification for type of exercise intervention.

### Sensitivity analysis: investigation of risk of bias

Effect estimates will be explored using data only from trials deemed to be at low risk of bias from: random sequence generation; allocation concealment; blinded outcome assessment; incomplete outcome data; and trials deemed to be at low risk of bias across all domains.

### Unavailable IPD

For trials where IPD were not obtained, we will seek to extract suitable aggregate data from their trial publications and combine these with the IPD trials using suitable statistical methods, to determine whether conclusions remain the same.[27] This is only likely to be possible when examining overall effects, as subgroup differences (interactions) are rarely reported.

### Investigation of small study effects

In meta-analyses of 10 trials or more, contour-enhanced funnel plots and tests for asymmetry will be used to investigate small trial effects and the potential for publication bias. Restriction to 10 trials is because there is low power to detect small trial effects with few studies.[52]

## ETHICS AND DISSEMINATION

Research ethical or governance approval is exempt for this study as no new data are being collected.[53 54] Findings will be presented at national (UK) and international conferences and submitted for publication in high-quality peer review journals. We will more broadly disseminate the results to physiotherapists, GPs, practice nurses, orthopaedic surgeons, patients and the general public both nationally and internationally by posting summaries on university websites, placing summaries in local healthcare settings and sending a summary to relevant groups and organisations for wider dissemination, for example, Osteoarthritis Research Society International, the European League Against Rheumatism, the American College of Rheumatology and Arthritis Research UK. Our PPIE working group will assist in developing plain English summaries and will jointly write articles that will be sent to newspapers, news websites, radio and other news media for wider dissemination.

**Author affiliations**
[1]Arthritis Research UK Primary Care Centre, Research Institute of Primary Care and Health Sciences, Keele University, Keele, UK
[2]Department of General Practice, Erasmus University Medical Center Rotterdam, Rotterdam, The Netherlands
[3]Department of Surgical Sciences, Centre for Musculoskeletal Outcomes Research, Orthopaedic Surgery Section, Dunedin School of Medicine, University of Otago, Dunedin, New Zealand

[4]Sargent College, Boston University, Boston, Massachusetts, USA
[5]Research User Group, Arthritis Research UK Primary Care Centre, Research Institute of Primary Care and Health Sciences, Keele University, Keele, UK
[6]Department of Physiotherapy, Centre for Health, Exercise & Sports Medicine, University of Melbourne, Melbourne, Victoria, Australia
[7]Faculty of Health, ACHIEVE Centre of Expertise, Amsterdam University of Applied Sciences, Amsterdam, The Netherlands
[8]Coronel Institute of Occupational Health, Academic Medical Center, University of Amsterdam, Amsterdam, The Netherlands
[9]Faculty of Health Sciences, School of Rehabilitation Sciences, University of Ottawa, Ottawa, Canada
[10]Division of Physical Therapy, Faculty of Health Science, Srinakharinwirot University, Bangkok, Thailand
[11]Musculoskeletal Statistics Unit, The Parker Institute, Frederiksberg and Bispebjerg Hospital, Copenhagen, Denmark
[12]Centre for Research Action in Public Health, University of Canberra, Bruce, Australian Capital Territory, Australia
[13]Amsterdam Rehabilitation Research Centre, Centre for Rehabilitation and Rheumatology, Reade, Amsterdam, The Netherlands
[14]Academic Rheumatology, University of Nottingham, Nottingham, Nottinghamshire, UK
[15]School of Physiotherapy, Royal College of Surgeons in Ireland, Dublin, Ireland
[16]TNO Netherlands Organisation for Applied Scientific Research, Leiden, The Netherlands
[17]Department of Public and Occupational Health, EMGO Institute for Health and Care Research, VU University Medical Center, Amsterdam, The Netherlands
[18]Faculty of Health, Social Care and Education, St George's University of London and Kingston University, London, UK
[19]Health Innovation Network South London, London, UK
[20]Department of Sports Medicine, Medical Clinic, University Hospital of Tübingen, Tübingen, Germany
[21]Manchester Movement Unit, Manchester School of Physiotherapy, Manchester Metropolitan University, Manchester, UK
[22]J.B. Snow Biomechanics Laboratory, Department of Health and Exercise Science, Worrell Professional Center, Wake Forest University, Winston Salem, USA
[23]Department of Health Services, University of Washington, Seattle, Washington, USA
[24]Department of Physical Medicine and Rehabilitation, Medical Faculty, Balikesir University, Balikesir, Turkey
[25]Department of Neurology, University of Tennessee Health Science Center, College of Medicine, Memphis, Tennessee, USA

**Contributors** MAH, DLB, JR, DvdW, RDR, KD, NEF contributed to the initial conception of the study. MAH, DLB, JR, DvdW, RDR, KD, AL, ALE, HA, KB, JB, KLB, DB, LB, KC, RC, TC, MdR, MD,HPF, SH, RSH, MHR, MVH, CI, JK, IK, CM, SPM, DLP, NS, LAT, RT, CHT, MvM, CW and NEF made a substantial contribution to the design of the work. MAH, DLB and JR drafted the manuscript. MAH, DLB, JR, DvdW, RDR, KD, AL, ALE, HA, KB, JB, KLB, DB, LB, KC, RC, TC, MdR, MD,HPF, SH, RSH, MHR, MVH, CI, JK, IK, CM, SPM, DLP, NS, LAT, RT, CHT, MvM, CW and NEF revised the manuscript and approved the final submission. The OA Trial Bank steering committee peer reviewed and approved the study protocol. The guarantor of the review is NEF.

**Funding** This work is supported by a Grant from the Chartered Society of Physiotherapy Charitable Trust (grant no PRF/16/A07), and the National Institute for Health Research (NIHR) School of Primary Care Research (grant no 531). MAH was supported by a National Institute for Health Research (NIHR) School of Primary Care Research Fellowship. DLB is currently supported by a NIHR School of Primary Care Research Fellowship. JR received partial funding from a grant of the Dutch Arthritis Foundation for their center of excellence 'osteoarthritis in primary care'. KB is funded by an Australian National Health and Medical Research Council Fellowship (no 1058440). Musculoskeletal Statistics Unit, The Parker Institute, Bispebjerg and Frederiksberg Hospital (RC) is supported by a core grant from the Oak Foundation (OCAY-13-309). DvdW is a member of PROGRESS Medical Research Council Prognosis Research Strategy (PROGRESS) Partnership (G0902393/99558). KD is part-funded by a Knowledge Mobilisation Research Fellowship (KMRF-2014-03-002) from the NIHR and the NIHR Collaborations for Leadership in Applied Health Research and Care West Midlands. RSH is supported by an Australian Research Council Future Fellowship (FT130100175). NEF, a NIHR Senior Investigator, is supported through an NIHR Research Professorship (NIHR-RP-011-015). The funders did not influence the study design or the writing of this article.

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
