## [Reviewer comments · BMJ Open]

ARTICLE DETAILS

TITLE (PROVISIONAL)	Subgrouping and TargetEd Exercise pRogrammes for knee and hip OsteoArthritis (STEER OA): A systematic review update and individual participant data meta-analysis protocol
AUTHORS	Holden, Melanie; Burke, Danielle; Runhaar, Jos; van der Windt, Danielle; Riley, Richard; Dziedzic, Krysia; Legha, Amardeep; Evans, Amy; Abbott, J. Haxby; Baker, Kristin; Brown, Jenny; Bennell, Kim; Bossen, Daniël; Brosseau, Lucie; Chaipinyo, Kanda; Christensen, Robin; Cochrane, Tom; de Rooij, Mariette; Doherty, Michael; French, Helen; Hickson, Sheila; Hinman, Rana S.; Hopman-Rock, M; Hurley, Mike; Ingram, Carol; Knoop, J; Krauss, Inga; McCarthy, Chris; Messier, Stephen; Patrick, Donald; Sahin, Nilay; Talbot, Laura; Taylor, Robert; Teirlinck, Carolien; van Middelkoop, Marienke; Walker, Christine; Foster, Nadine

VERSION 1 – REVIEW

REVIEWER	Nina Østerås Diakonhjemmet Hospital, Norway
REVIEW RETURNED	28-Aug-2017

GENERAL COMMENTS	The manuscript is a protocol for a planned systematic review update and individual participant data meta-analysis targeting therapeutic exercise for knee and hip osteoarthritis. The study aims to identify subgroups that do/ do not respond to therapeutic exercise and to different types of exercise, and to explore mediators of the effect of therapeutic exercise. Knowledge on optimal targeting and refining the content of exercise interventions is highly warranted. The protocol is well written and the methods and analyses are clearly described. I have only a few minor comments and suggestions. Minor comments: Phase 1, Study selection and Table 1 1. Page 13, line 286: How do you define “adults”. Often this is >18 years old, but in Table 1 “mean age >45 is mentioned”. Consider to include an exact lower age limit.2. Page 13-14 and Table 1: The text on page 13-14 on Study selection has a large overlap (repetition), but is not identical to the text in Table 1 Inclusion/exclusion criteria. For instance on page 13 the study population is defines as “adults”, whereas in Table 1 “aged 45 years and over” is mentioned. On page 13 regarding the Comparator, examples of “no exercise control group” is provided, but not for “other forms of exercise”, but Table 1 include both set of examples.3. Study selection: In which category would you put passive interventions (i.e manual therapy)? Would “sham” ultrasound be categorized as placebo/attention control?
---

	4. Table 1, page 15, lines 11-15: Suggest to slightly revise the sentence: “Exercise or physical activity....not specifically applied to improve or maintain knee and/or hip OA”, as I believe exercise therapy aims to improve or maintain symptom and function levels – not to maintain OA... 5. Table 1, page 15, lines 43-44: Are measures of both pain and physical function required, or could a study with only pain measures, but not function, be included? Is it only “and” or could it be “and/or”? Extraction of aggregate data 6. Page 16: In some studies the participants are followed after the intervention period has ended, and the effects of exercise will normally fade out if the participants discontinue their exercise programme. Will you also extract data at follow-up time points long after the intervention period has ended? This is only mentioned related to analyses of Objective 6 (page 34, line 499). 7. Page 16, line 312: Suggest to include a specific definition of low, moderate and high intensity 8. Page 16-17: Will you extract and use/analyze data regarding the participants’ adherence or compliance to the therapeutic exercise intervention? Part 3: Statistical Analyses 9. Page 21, line 419: “...a considerable amount of missing data...” How much is “considerable”? Suggest including a more precise statement. 10. Page 21-22: Do I understand you correctly in that studies comparing different types of exercise will not be included in analyses related to Objective 1 and 3 (exercise vs. no exercise), but included in analyses related to Objective 2 and 4?
--	---

REVIEWER	George Ampat Royal Liverpool University Hospital, Liverpool, UK No Competing Interest
REVIEW RETURNED	03-Sep-2017
GENERAL COMMENTS	Probes a very relevant part of the disease of OA for which an answer is still not known.

VERSION 1 – AUTHOR RESPONSE

Reviewer 1

Comment 1. Page 13, line 286: How do you define “adults”? Often this is >18 years old, but in Table 1 “mean age >45 is mentioned”. Consider to include an exact lower age limit.

Response: In line with the NICE OA guidelines, for a clinical diagnosis of OA, a lower age limit of 45 years is recommended (1). Thus, our age limit is adults aged 45 years and over. As shown below, for clarity, we have specified in the text that the adult population refers to study populations of age 45 years and above (page 13). This is now consistent with Table 1.

“Study population: Adults age 45 years and above with knee or hip OA...;”

Comment 2. Page 13-14 and Table 1: The text on page 13-14 on Study selection has a large overlap (repetition), but is not identical to the text in Table 1 Inclusion/exclusion criteria. For instance on page 13 the study population is defines as “adults”, whereas in Table 1 “aged 45 years and over” is mentioned. On page 13 regarding the Comparator, examples of “no exercise control group” is provided, but not for “other forms of exercise”, but Table 1 include both set of examples.

Response: As shown below, we have added text (page 13) to explain that the full details of the study selection criteria are listed in Table 1, with a briefer summary provided in the body of the manuscript. This is to avoid repetition with Table 1 and for brevity in the body of the text. For consistency, examples of “no exercise control group” have been removed from the body of the text; all examples can only be found in Table 1 (page 15).

“Study Selection

Full details of the study selection criteria are listed in Table 1. In summary, we will evaluate RCTs against the following inclusion criteria:

Study population: Adults age 45 years and above with knee or hip OA; Intervention: Any land or water based therapeutic exercise intervention regardless of content, duration, frequency, or intensity;

Comparator: Other forms of exercise or no exercise control group; Outcome measure: At least....”

Comment 3. Study selection: In which category would you put passive interventions (i.e. manual therapy)? Would “sham” ultrasound be categorized as placebo/attention control?

Response: Other ‘active’ treatments as exclusions would include manual therapy, however we now realise this may be interpreted to mean physically active treatments. For clarity we have removed the word ‘active’ in Table 1, and given manual therapy as an example. This now reads as follows (page 15):

(Comparator exclusion)

“If the comparator is a different intervention (for example manual therapy, ultrasound, intra-articular injection, opioids, weight loss, etc), other than usual care, waiting list, placebo, attention control, or no treatment”.

Sham treatment (e.g. sham ultrasound) has been added to the list of inclusion criteria for Comparators in Table 1 (page 15).

Comment 4. Table 1, page 15, lines 11-15: Suggest to slightly revise the sentence: “Exercise or physical activity....not specifically applied to improve or maintain knee and/or hip OA”, as I believe exercise therapy aims to improve or maintain symptom and function levels – not to maintain OA...

Response: As suggested, we have amended the sentence to:
“Exercise or physical activity that was not specifically applied to improve OA symptoms and function” (Table 1, page 15).

Comment 5. Table 1, page 15, lines 43-44: Are measures of both pain and physical function required, or could a study with only pain measures, but not function, be included? Is it only “and” or could it be “and/or”?

Response: Studies are not required to include measures of both pain and function. For clarity, we have therefore amended the exclusion criteria text to state:
“No measure of self-reported pain and/or physical function” (Table 1, page 15).

Comment 6. Page 16: In some studies the participants are followed after the intervention period has ended, and the effects of exercise will normally fade out if the participants discontinue their exercise programme. Will you also extract data at follow-up time points long after the intervention period has ended? This is only mentioned related to analyses of Objective 6 (page 34, line 499).

Response: We will extract outcome data on self-report pain and/or physical function at time points nearest to 12 weeks, 6 months and 12 months post randomisation, regardless of the timing of the intervention. At these follow up time points, some participants will have completed their intervention, while in other studies the intervention may be ongoing. We will explore the influence of duration of the exercise program as a study level variable in addressing objective two. As shown below, we have now specified in the text that follow up time points are post randomisation for clarity.

“Two reviewers will independently extract outcome data on self-report pain and/or physical function at time points nearest to 12 weeks, 6 months and 12 months post randomisation.” (page 17).

“All self-report pain and physical function outcome data at time-points nearest to 12 weeks (short-term), 6 months (mid-term) and 1 year (long-term) post randomisation.” (page 20).

“We will attempt to re-produce the results included in each initial trial publication, including baseline characteristics and self-reported pain and physical function at a time point nearest to 12 weeks, 6 months and 1 year post randomisation.” (page 20).

Comment 7. Page 16, line 312: Suggest to include a specific definition of low, moderate and high intensity

Response: In line with previous literature (2,3), a definition of intensity has now been specified in the text. This now reads as follows (page 17):

Intensity of exercise: Low, moderate or high intensity (based on published information regarding target heart rate (<50% of maximum heart rate (MHR) = low intensity, 50–70% MHR = moderate intensity, >70–85% MHR = high intensity) or Metabolic Equivalent (MET) score (where heart rate information is unavailable) (MET score of <3 = low intensity, MET 3-6 = moderate intensity, MET > 6 = high intensity [34])) [35]; low or high impact (categorised based on the likely amount of compressive load and whether both feet were intermittently off the ground. For example, cycling, swimming and walking = low impact; jogging, running and jumping = high impact) [35].

Comment 8. Page 16-17: Will you extract and use/analyze data regarding the participants' adherence or compliance to the therapeutic exercise intervention?

Response: The collaborators on this project have also identified that this is an important topic area to explore, which warrants a separate analysis. The meta-analysis proposed here will focus on baseline characteristics of individual patients, study-level characteristics of interventions, and a limited number of potential mediators. Adherence to exercise is a complex construct that has been measured in many different ways, and in some interventions will be specifically addressed. Rather than incorporating this in an already large meta-analysis, we have proposed to address this in a separate project.

Comment 9. Page 21, line 419: "...a considerable amount of missing data..." How much is "considerable"? Suggest including a more precise statement.

Response: The text has been amended to include a definition of considerable:
"If there is a considerable amount of missing baseline data (> 5% of patients have one or more missing values) for particular variables of interest (such as potential individual-level effect moderators) this will be handled using within-study multiple imputation..." (page 22)

Comment 10. Page 21-22: Do I understand you correctly in that studies comparing different types of exercise will not be included in analyses related to Objective 1 and 3 (exercise vs. no exercise), but included in analyses related to Objective 2 and 4?

Response: The authors confirm this is correct. Objectives 1 and 3 can only be achieved in studies where the effect of therapeutic exercise is determined against a non-exercise control. In objectives 2 and 4, we will consider all forms of therapeutic exercise used in all included studies.

References:

1. National Institute for Health and Care Excellence. Osteoarthritis: care and management. Clinical guideline [CG177]. 2014 [Available from: <https://www.nice.org.uk/guidance/cg177>].
2. Ainsworth BE, Haskell WL, Herrmann SD, et al. Compendium of physical activities: a second update of codes and MET values. *Med Sci Sports Exerc* 2011;43(8):1575-1581.
3. Quicke JG, Foster NE, Thomas MJ, et al. Is long-term physical activity safe for older adults with knee pain?: a systematic review. *Osteoarthritis Cartilage* 2015;23(9):1445-56.

VERSION 2 – REVIEW

REVIEWER	Nina Østerås Diakonhjemmet Hospital, Norway
REVIEW RETURNED	19-Oct-2017
GENERAL COMMENTS	The authors have responded satisfactorily to all my comments and revised the protocol accordingly. Good luck with this important study.